# Bacterial outer membrane vesicles suppress tumor by interferon-γ-mediated antitumor response

Oh Youn Kim [1], Hyun Taek Park[1], Nhung Thi Hong Dinh[1], Seng Jin Choi[1], Jaewook Lee[1], Ji Hyun Kim[1], Seung-Woo Lee[1] & Yong Song Gho [1]

Gram-negative bacteria actively secrete outer membrane vesicles, spherical nano-meter-sized proteolipids enriched with outer membrane proteins, to the surroundings. Outer membrane vesicles have gained wide interests as non-living complex vaccines or delivery vehicles. However, no study has used outer membrane vesicles in treating cancer thus far. Here we investigate the potential of bacterial outer membrane vesicles as therapeutic agents to treat cancer via immunotherapy. Our results show remarkable capability of bacterial outer membrane vesicles to effectively induce long-term antitumor immune responses that can fully eradicate established tumors without notable adverse effects. Moreover, systematically administered bacterial outer membrane vesicles specifically target and accumulate in the tumor tissue, and subsequently induce the production of antitumor cytokines CXCL10 and interferon-γ. This antitumor effect is interferon-γ dependent, as interferon-γ-deficient mice could not induce such outer membrane vesicle-mediated immune response. Together, our results herein demonstrate the potential of bacterial outer membrane vesicles as effective immunotherapeutic agent that can treat various cancers without apparent adverse effects.

[1] Department of Life Sciences, Pohang University of Science and Technology, Pohang 37673, Republic of Korea. Oh Youn Kim, Hyun Taek Park and Nhung Thi Hong Dinh contributed equally to this work. Correspondence and requests for materials should be addressed to O.Y.K. (email: aglaia@postech.ac.kr) or to Y.S.G. (email: ysgho@postech.ac.kr)

Cancer immunotherapies such as the immune checkpoint blockade therapies and chimeric antigen receptor therapies are now being recognized as a promising approach to overcome various cancers[1–9]. Although many studies have focused on combining nanotechnology with chemotherapy for delivery of chemotherapeutics, immunotherapies using nanoparticles have only been minimally explored. However, nanosized particles can easily flow through the blood and lymphatic vessels, and can readily interact with or be ingested by immune cells, giving them great potential as immunostimulatory agents[10–12]. Bacterial outer membrane vesicles (OMVs), also known as extracellular vesicles, are naturally produced from all Gram-

negative bacteria and have nano-sized lipid-bilayered vesicular structures composed of various immunostimulatory components[13–16]. This acellular bacterial OMV is one of the cutting-edge immunostimulatory agents recognized by many scientists as candidate vaccines and delivery vehicles[17–19]. However, despite such advantages of bacterial OMVs as cancer immunotherapeutic agent, no study has yet examined the potential of bacterial OMVs in treating cancer. We have cited the supplementary methods in the manuscript text (in five places).

One of the important aspects that need to be addressed in cancer immunotherapy is the toxicity issue. Although the history of immunotherapy dates back to the early 1890s when Dr

**Fig. 1** Treatment of outer membrane vesicles (OMVs) induces complete regression of tumors. **a** Transmission electron micrograph image of *E. coli* W3110 wild type-derived OMVs (WT OMVs) and *E. coli* W3110 *msbB* mutant-derived OMVs (Δ*msbB* OMVs). *Scale bars*, 100 nm. **b** Size distribution of *E. coli* W3110 WT and Δ*msbB* OMVs measured by dynamic light scattering analysis ($n = 5$). **c** Production yield of *E. coli* W3110 WT and Δ*msbB* OMVs from $1 \times 10^9$ CFUs bacteria in terms of total protein amount ($n = 5$, three independent experiments). **d** Tumor volume of mice bearing CT26 murine colon adenocarcinoma measured after *E. coli* Δ*msbB* OMV treatments with various amounts (total $n = 14$ mice per group, two independent experiments). *E. coli* Δ*msbB* OMVs were injected intravenously four times from day 6 with 3 days intervals. **e** Picture of tumor and tumor tissue histology 48 h after single PBS or Δ*msbB* OMV (5 μg in total protein amount) injection. *Scale bars*, 50 μm. **f** Picture of mice bearing tumor after PBS or Δ*msbB* OMV (5 μg in total protein amount) treatments. *Yellow box* indicates tumor sites. **g** Mice treated with OMV (5 μg in total protein amount) with complete tumor regression were re-challenged with tumors on the opposite flank of the mice 4 weeks after the final injection. Then, these mice were re-challenged with tumor on the middle of the two flanks of the mice 3 weeks after the secondary tumor injection (total $n = 14$ mice per group, two independent experiments). Data are presented as the mean ± SD from a representative experiments. \*\*\*$P < 0.001$ analysed by unpaired Student's $t$-test **c** or two-way ANOVA with Bonferroni post test to compare each treated group with PBS group **d**, **g**

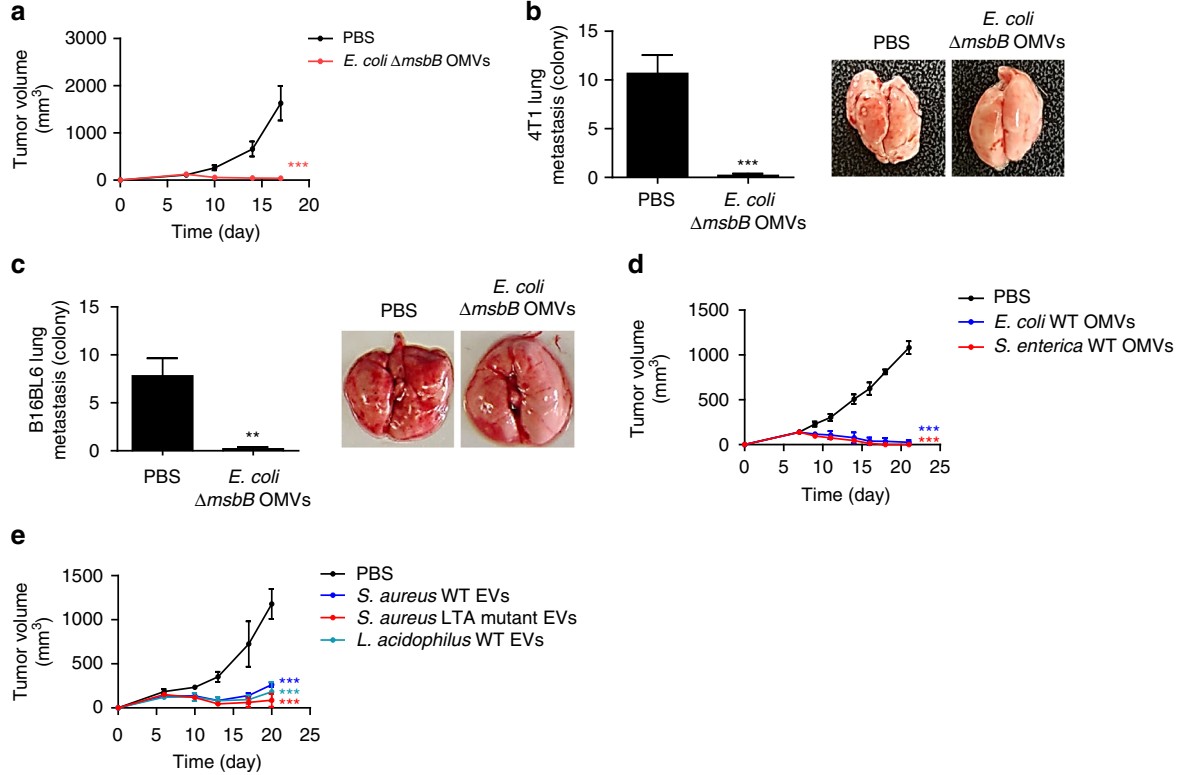

**Fig. 2** Systemic administration of bacterial extracellular vesicles induces effective antitumor responses in multiple tumors. **a** Tumor volume of mice measured after the subcutaneous injection of MC38 murine adenocarcinoma (total $n = 14$ mice per group, two independent experiments). *E. coli* Δ*msbB* OMVs (5 μg in total protein amount) were injected intravenously four times from day 7 with 3 days intervals. **b**, **c** To induce spontaneous lung metastasis, highly metastatic 4T1 murine carcinoma cells and B16BL6 melanoma cells were subcutaneously injected to the right flank of the mice (total $n = 12$ mice per group, two independent experiments). PBS or *E. coli* Δ*msbB* OMVs (5 μg in total protein amount) were intravenously treated for four times with 3 days intervals from day 7. At day 22, mice were killed and the number of colonies spontaneously metastasized in the lungs for 4T1 **b** and B16BL6 **c** tumors were counted. **d**, **e** Tumor volume of mice bearing CT26 tumor of mice intravenously injected with four doses of 5 μg of Gram-negative *E. coli* W3110 wild type- and *Salmonella enterica* wild type-derived OMVs. **d** Gram-positive bacteria *Staphylococcus aureus* wild type- and lipoteichoic acid (LTA) mutant-derived extracellular vesicles (EVs), and *Lactobacillus acidophilus* wild type-derived extracellular vesicles (EVs), **e** ($n = 5$ mice per group). Data are presented as the mean ± SD from a representative experiments. *$P < 0.01$ and **$P < 0.001$, respectively, analysed by unpaired Student's *t*-test **b**, **c** or two-way ANOVA **a**, **d**, **e**. Bonferroni post test was applied to compare each treated group with PBS group **d**, **e**

William Coley used a mixture of weakened bacteria solution as Coley's toxin to cure cancer patients, only recently has this concept of immunotherapy been investigated extensively due to safety concerns[20, 21]. Compared with live or weakened bacteria, OMVs are considered safe, as they are acellular and effective in small quantity. In fact, OMV-based vaccine is being clinically used as meningococcal group B vaccine in children under the trade name Bexsero[13, 22].

Here in this study, we examined whether bacterial OMVs could be employed as therapeutic agents to treat cancer via immunotherapy. We found that bacterial OMVs have significant ability to effectively induce antitumor immune responses and fully eradicate established tumors without notable adverse effects. This antitumor response of bacterial OMVs is durable, and secondary and tertiary re-challenges of tumor are fully rejected by mice that were cured from primary challenge. In addition, systematically administered bacterial OMVs target and accumulate in the tumor tissue, and subsequently induce the production of antitumor cytokines CXCL10 and interferon (IFN)-γ. Moreover, IFN-γ-deficient mice do not induce such bacterial OMV-mediated immune response, suggesting that this antitumor effect is dependent on IFN-γ. Together, the results in this study demonstrate the great potential of bacterial OMVs as novel therapeutic agents that can trigger effective antitumor response against various cancers without notable adverse effects.

## Results

**Generation and characterization of bacterial OMVs.** Bacterial OMVs can easily be modified through genetic engineering. In this study, to avoid possible adverse effects due to bacterial endotoxin lipopolysaccharide, we used Gram-negative bacterial OMVs derived from genetically modified *Escherichia coli*, whose gene encoding lipid A acyltransferase (*msbB*), the lipid component of lipopolysaccharide, had been inactivated (*E. coli* msbB$^{-/-}$, Δ*msbB*)[23–25]. To test whether our Δ*msbB* mutant bacteria-derived OMVs have impaired lipid A and therefore does not react to TLR4 receptor responsible for recognizing lipopolysaccharide, we treated *E. coli* wild type and Δ*msbB* OMVs to TLR4/MD2-transfected human embryonic kidney HEK293 cells (Supplementary Fig. 1 and Supplementary Methods). As a result, Δ*msbB* OMV-treated cells did not produce any interleukin (IL)-8 cytokines, whereas *E. coli* wild-type OMV-treated cells produced high levels of IL-8 cytokines, suggesting Δ*msbB* OMVs have impaired lipid A. Next, we carried out characterization experiments using *E. coli* W3110 wild type and Δ*msbB* mutant bacteria-derived OMVs, to examine the physiological traits of OMVs. Transmission electron micrograph images and dynamic light scattering analyses showed that both *E. coli* wild type and Δ*msbB* mutant bacteria-derived OMVs have nano-sized lipid-bilayered vesicular structures with the average diameters of 38.6 ± 3.6 and 38.7 ± 4.2 nm, respectively (Fig. 1a, b). To our surprise, *E. coli* Δ*msbB*

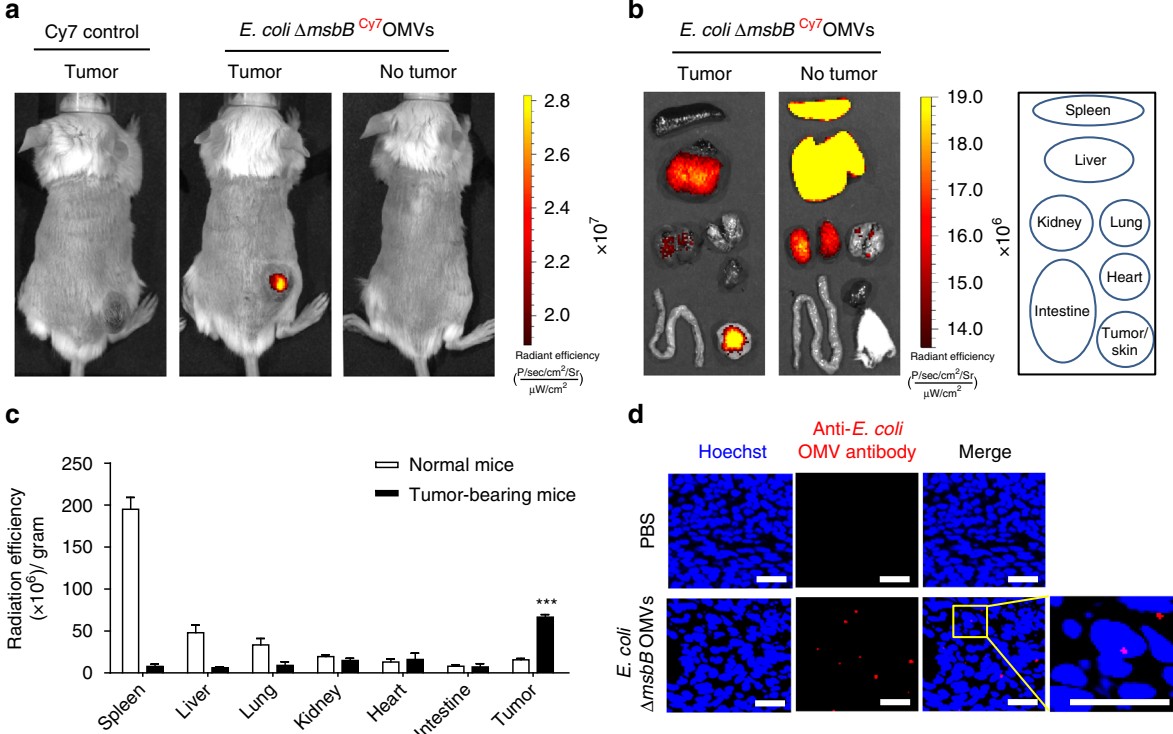

**Fig. 3** Targeting of *E. coli* Δ*msbB* OMVs to tumor tissues after systemic administration. **a** Cy7 control and Cy7-labeled *E. coli* Δ*msbB* OMVs (Δ*msbB* Cy7OMVs) were systematically injected to BALB/c mice bearing CT26 tumor cells. For control, Δ*msbB* Cy7OMVs were also injected to healthy BALB/c mice with no tumor. Whole body distributions of the injected Cy7OMVs were observed using in vivo imaging system spectrum 12 h after the injection. **b** Spleen, liver, kidney, lung, heart, intestine, and tumor tissues were isolated to measure the accumulation of Cy7 fluorescence in different organs. **c** Radiant efficiencies of each organ were acquired for Cy7 fluorescence using Living Image 3.1 Software and were normalized by each organ weight. Results are from three independent experiments (total *n* = 9). **d** For tumor tissue immunohistochemistry, Δ*msbB* OMVs were intravenously injected to BALB/c mice bearing CT26 tumor. Tumor tissues were extracted after 12 h and were embedded in paraffin for tissue section analyses by immunohistochemistry. The cell nucleus is stained in blue (Hoechst) and OMVs are shown in red fluorescence signal (anti-*E. coli* OMV polyclonal antibodies)[36] Scale bars, 50 μm. ***P < 0.001 analysed by unpaired Student's *t*-test

OMVs had higher production yield compared with wild-type *E. coli* OMVs, providing further advantage of using *E. coli* Δ*msbB* OMVs, as naturally derived OMVs have concerns regarding low productivity[18] (Fig. 1c).

**Antitumor response induced by systemic injection of OMVs.** Initially, to investigate the antitumor effect of Gram-negative bacterial OMVs, we intravenously administered *E. coli* W3110 Δ*msbB* OMVs of varying amounts to mice subcutaneously transplanted with CT26 murine colon adenocarcinoma (Fig. 1d). Treatments of Δ*msbB* OMVs caused significant reduction in the tumor volume dose dependently with complete elimination of tumor tissue for mice injected with 5 μg of Δ*msbB* OMVs. Systemic injections of certain anaerobic bacteria such as *Salmonella typhimurium* have previously been shown to induce antitumor effect by targeting to tumor tissue[26, 27]. To confirm whether this antitumor effect is solely induced by OMVs, we tested the antitumor effect of *E. coli* Δ*msbB* mutant bacteria (Supplementary Fig. 2). It is noteworthy that 5 μg of Δ*msbB* OMVs are produced from $1 \times 10^9$ colony-forming units (CFUs) *E. coli* Δ*msbB* bacteria. As a result, administration of *E. coli* Δ*msbB* bacteria could not induce antitumor response. Furthermore, all mice injected with $1 \times 10^9$ CFUs died within 48 h after the injection and most of the mice developed systemic inflammatory response syndrome symptoms such as the formation of eye exudates, piloerection or hypothermia[28, 29]. When we intravenously injected Δ*msbB* OMVs (5 μg) to mice bearing CT26 tumors, the mice did not show notable adverse effects and did not show change in body temperature, or body weight (Supplementary Fig. 3a, b).

Moreover, we also assessed the major organs such as the lungs, liver, spleen, and kidney for any organ damage by histological analyses but could not detect distinguished organ destruction (Supplementary Fig. 3c). However, in-depth safety evaluation of the OMV injections should be evaluated in the future to validate their safety for clinical use. On the other hand, distinct phenotypical and histological changes in the tumor tissue, such as the darkening of the tumor surface and increase in the necrotic and apoptotic areas, respectively, were observed 48 h after the first OMV injection (Fig. 1e). All the tumors in mice treated with 5 μg of Δ*msbB* OMVs were completely eliminated (Fig. 1f).

**Long-term memory effect of OMV-induced antitumor response.** Next, to investigate whether this antitumor response involved effective immunological memory response, we re-challenged CT26 cells to mice treated with 5 μg of Δ*msbB* OMVs, who completely recovered from the first CT26 challenge. The secondary tumor was subcutaneously injected to the opposite flank of the primary tumor challenged site 4 weeks after the last OMV treatment (Fig. 1g). All the mice treated with OMVs rejected the secondary challenge of CT26 cells. Moreover, these OMV-treated mice also rejected the tertiary challenge of the CT26 tumor on the middle of the two flanks 3 weeks after the secondary challenge.

**Universality of extracellular vesicle-induced antitumor response.** To verify whether this antitumor activity of OMVs is a general phenomenon applicable to various cancer or OMV types,

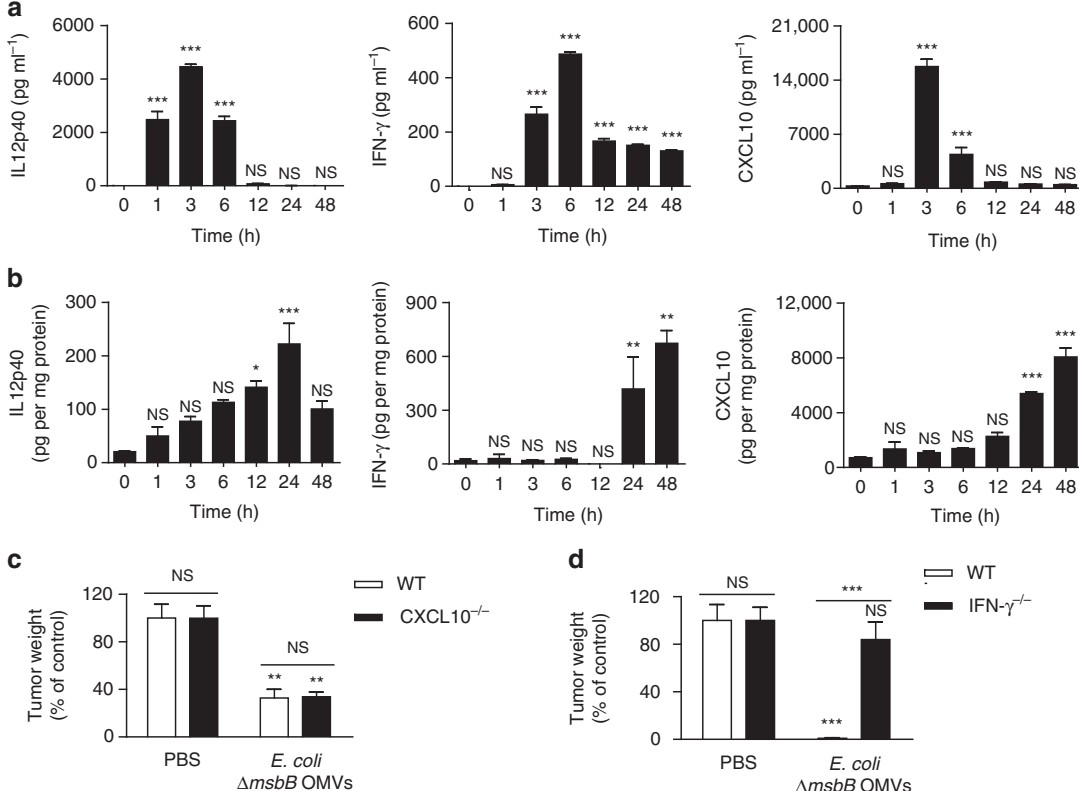

**Fig. 4** *E. coli ΔmsbB* OMV antitumor effect is IFN-γ dependent. **a**, **b** Release of antitumor cytokines IL-12p40, IFN-γ, and CXCL10 in blood sera **a** and tumor cell lysate **b** after single intravenous injection of *E. coli ΔmsbB* OMVs (5 μg in total protein amount) to mice bearing CT26 tumors at different time points. **c**, **d** The antitumor efficacy of OMV treatment in CXCL10-deficient (CXCL10$^{-/-}$) mice **c** and IFN-γ-deficient (IFN-γ$^{-/-}$) mice **d**. Data are shown as the mean ± SD ($n = 6$ mice per group). NS, not significant, *$P < 0.05$, **$P < 0.01$, and ***$P < 0.001$, respectively, analysed by one-way ANOVA. Bonferroni multiple comparisons post test was applied to compare each time point to zero time point **a**, **b** or to compare treated group with PBS group of each mouse **c**, **d**

we performed the same antitumor experiments on different tumors or with different OMV strains. First, when *E. coli ΔmsbB* OMVs were administered to mice having different genetic background and were challenged with MC38 colon cancer cells, strong antitumor activity was observed (Fig. 2a). In addition, intravenous injections of OMVs to mice subcutaneously transplanted with highly metastatic 4T1 murine carcinoma and B16BL6 melanoma cells showed complete prevention of spontaneous metastasis in the lungs (Fig. 2b, c), as well as significant reduction in the primary tumor growth (Supplementary Fig. 4), although not as fully effective as the previous antitumor activity shown for CT26 colon adenocarcinoma models (Fig. 1d). This OMV-mediated antitumor effect was also observed when treated with OMVs from different Gram-negative bacterial strains (Fig. 2d) and extracellular vesicles derived from Gram-positive origins to mice bearing CT26 tumors (Fig. 2e and Supplementary Fig. 5). Interestingly, bacterial extracellular vesicles derived from *Lactobacillus acidophilus*, a foodborne Gram-positive bacterial strain, also showed significant antitumor effect, suggesting the potential of using bacterial extracellular vesicles derived from good bacteria in future clinical applications. In addition, to check whether the tumor regression effect is maintained for long term without tumor rebound, we monitored the tumor volume of *Staphylococcus aureus* wild-type extracellular vesicle-treated mice for long term. We did not observe any tumor volume rebound even after 5 weeks of final *S. aureus* extracellular vesicle treatment (Supplementary Fig. 6).

**In vivo distribution and targeting ability of bacterial OMVs.** Next, we sought to determine the underlying mechanism of OMV-induced antitumor effect. First, we labeled *E. coli ΔmsbB*

OMVs with Cy7 fluorescence (ΔmsbB $^{Cy7}$OMVs) and tracked their distribution after systemic administration through in vivo imaging system. After 12 h of the intravenous administration of ΔmsbB $^{Cy7}$OMVs to mice bearing CT26 tumor or normal mice, we measured the fluorescence intensity of Cy7 in whole body (Fig. 3a) and in different organs including the spleen, liver, kidney, lung, heart, intestine, and tumor (Fig. 3b and Supplementary Fig. 7). Strong fluorescence signal was observed in the tumor tissue of the mice injected with $^{Cy7}$OMVs. When the radiation efficiency of Cy7 signal was divided by each organ weight, tumor tissue had the highest intensity in tumor-bearing mice, suggesting accumulation of *E. coli ΔmsbB* OMVs on tumor tissue (Fig. 3c). In addition, when Cy7-labeled *S. aureus* extracellular vesicles were intravenously injected to mice bearing tumors, *S. aureus* extracellular vesicles were accumulated in the tumor tissue (Supplementary Fig. 8 and Supplementary Methods). Moreover, we also performed immunohistochemistry analyses on tumor tissue sections 12 h after the systemic injection of OMVs, to confirm the infiltration of the OMVs in tumor tissue (Fig. 3d). This tumor-targeting ability of bacterial extracellular vesicles may be due to the passive targeting of the nano-sized bacterial extracellular vesicles to leaky tumor vasculature via enhanced permeability and retention, the EPR effect[30].

**Mechanism studies of bacterial OMV-induced antitumor effect.** We further performed additional experiments to find out the mode of action for OMV-induced antitumor response. As OMV antitumor effect is an early symptom, we measure the kinetic quantification of cytokines in blood serum and tumor tissue until 48 h after the intravenous injection of *E. coli ΔmsbB*

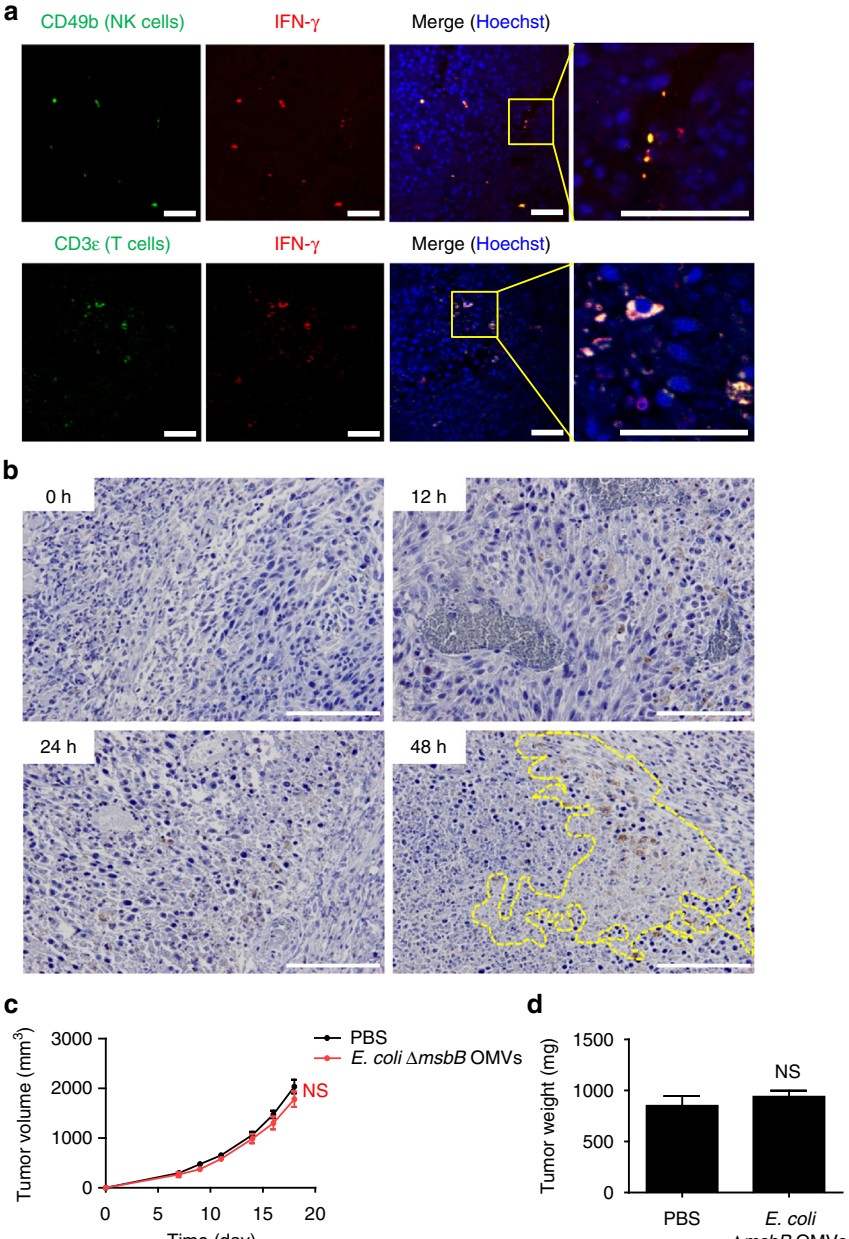

**Fig. 5** Importance of NK and T cells on OMV antitumor effect. **a** Images of tumor tissues isolated from wild-type mice bearing CT26 tumors, stained for IFN-γ, and NK cells (*top*) or T cells (*bottom*) 48 h after intravenous injections of *E. coli* Δ*msbB* OMVs. The cell nucleus is stained in *blue* (Hoechst), whereas NK and T cells are shown by *green* fluorescence signal and IFN-γ is shown in *red* fluorescence signal, respectively. *Scale bars*, 50 μm. **b** Images of tumor tissues isolated from wild-type mice bearing CT26 tumors, stained for NK cells (brown spots) at different time points after intravenous injections of *E. coli* Δ*msbB* OMVs. Tumor necrotic area around NK cells at 48 h is shown in dashed lines. *Scale bars*, 50 μm. **c**, **d** Tumor volume of NIHS-Lyst$^{bg}$Foxn1$^{nu}$Btk$^{xid}$ mice bearing CT26 tumor measured after *E. coli* Δ*msbB* OMV treatments **c** and tumor weight at the end of the experiment **d**. Data are presented as the mean ± SD ($n = 6$ mice per group). NS indicates not significant analysed by two-way ANOVA **c** and unpaired Student's *t*-test **d**

OMVs. We measured the cytokines and chemokines known to be involved in antitumor immune response such as the IL-12p40, IFN-γ, CXCL10, TNF-α, IL-6, and IL-12p70 (Fig. 4a, b and Supplementary Fig. 9)[31–33]. IFN-γ and CXCL10 increased time dependently in the tumor tissue (Fig. 4b). To assess the importance of these cytokines and chemokines in detail regarding OMV antitumor response, we performed the same antitumor experiment in mice deficient in CXCL10 (CXCL10$^{-/-}$) and IFN-γ (IFN-γ$^{-/-}$) (Fig. 4c, d). OMV treatments to mice with CXCL10 deficiency could also induce significantly effective antitumor response as with the wild-type mice (Fig. 4c and Supplementary Fig. 10a). However, OMV treatments to mice with IFN-γ deficiency could

not induce such antitumor effect (Fig. 4d and Supplementary Fig. 10b). In addition, we have performed additional neutralizing experiment with IFN-γ antibody to further verify that this cytokine is the main mechanism of OMV-induced antitumor response (see Supplementary Methods for details). Mice injected with mouse monoclonal anti-IFN-γ antibodies before OMV treatment did not show tumor regression, whereas mice injected with isotype IgG$_1$ antibodies before OMV treatment showed almost complete regression of tumor (Supplementary Fig. 11). Moreover, when Gram-positive *S. aureus* and *L. acidophilus* extracellular vesicles were treated to IFN-γ-deficient mice, antitumor response was not observed (Supplementary Fig. 12).

Together, this implies that IFN-γ has an important role in inducing both Gram-negative and Gram-positive bacterial extracellular vesicle-induced antitumor response.

In addition, IFN-γ was co-localized with natural killer (NK) and T cells in tumor tissues of mice 48 h after the OMV injection, suggesting that both NK and T cells produce IFN-γ after OMV injection (Fig. 5a and Supplementary Fig. 13). Moreover, in line with these results, we also observed that NK cells accumulate in the tumor necrotic area after OMV injection, especially at 48 h (Fig. 5b). To further support our data, we also carried out the antitumor experiments on mice deficient with the major producers of IFN-γ, the NK, and T cells (Fig. 5c, d)[34]. E. coli ΔmsbB OMV-induced antitumor response was not observed in NIHS-Lyst[bg]Foxn1[nu]Btk[xid] mice with deficiencies in cells responsible for IFN-γ production. Moreover, E. coli ΔmsbB OMV treatment to athymic nude (Nu/J (Fox1nu/Fox1nu), F120) mice with T-cell deficiency only showed about 50% of antitumor effect (Supplementary Fig. 14). Together with our previous results, this further provides evidence that IFN-γ has important roles in mediating E. coli ΔmsbB OMV-induced antitumor responses leading to tumor tissue disruption.

## Discussion

Gram-negative and Gram-positive bacterial extracellular vesicles are found in various biological fluids including the blood, urine, saliva, and feces, allowing new insight in the design of both personalized and universal cancer vaccines using commensal bacterial extracellular vesicles[35, 36]. In addition, being of bacterial origin, bacterial extracellular vesicles can be genetically modified to display various targeting moieties on the surface or load antigens as cargo without the complicated process of purifying or attaching targeting moieties or antigens[18, 19, 37, 38]. In our study, we only used genetic modifications to increase the safety of OMVs by removing bacterial endotoxin function. However, in the future, we could additionally modify the originating bacteria to produce OMVs expressing antibodies against cytotoxic T-lymphocyte-associated protein 4 and programmed cell death protein 1 for increased antitumor efficacy[39–41]. Furthermore, being of Gram-negative and Gram-positive bacterial origins, not only can bacterial extracellular vesicles be easily genetically modified to display various targeting moieties on the surface, but bacterial extracellular vesicles are also versatile containers that could be loaded with different types of chemotherapeutics and components including nucleic acids[37, 42]. This is especially appealing for future clinical studies in that bacterial extracellular vesicles, especially Gram-negative bacterial OMVs, can be applied to combination therapies to deliver various chemotherapeutics and induce antitumor responses at the same time to further augment therapeutic efficacy. However, for less destructive tumors such as CT26 and MC38 adenocarcinoma, bacterial OMV immunotherapy alone could be highly effective as monotherapy.

To investigate which components in the bacterial extracellular vesicles actually induce IFN-γ production, we performed additional experiments using extracellular vesicles of different condition (see Supplementary Methods for details). First, to examine whether the protein components of extracellular vesicles were the actual inducers of IFN-γ, we boiled both the E. coli ΔmsbB OMVs and S. aureus wild-type extracellular vesicles to denature protein structures. Then, we injected the heat-treated extracellular vesicles to tumor-bearing mice and measured the IFN-γ production in blood serum and tumor lysate (Supplementary Fig. 15a). As a result, IFN-γ production in both the blood serum and tumor tissue was not detected for heat-treated extracellular vesicles injected mice. Next, to further test whether the surface proteins of the extracellular vesicles are important in inducing IFN-γ

production, we treated trypsin to E. coli ΔmsbB OMVs and S. aureus wild-type extracellular vesicles to shave-off the vesicular surface proteins. Trypsin-treated extracellular vesicles also did not induce any IFN-γ production, suggesting that the surface proteins are the key factors involved in IFN-γ production (Supplementary Fig. 15b). Furthermore, we identified 200 and 476 vesicular proteins by the proteomic analyses of E. coli ΔmsbB OMVs and S. aureus wild-type extracellular vesicles, respectively (Supplementary Fig. 16, Supplementary Tables 1 and 2, and Supplementary Methods). We cultured E. coli ΔmsbB and S. aureus in lysogeny broth, which is composed of NaCl, yeast extracts, and tryptone (peptides formed by the digestion of trypsin of cow casein). Further analysis showed that neither yeast nor cow proteins were identified from proteomic analyses, suggesting that purified E. coli ΔmsbB OMVs and S. aureus wild-type extracellular vesicles are free of potential contaminants from the culture media. Taken together, these results suggest that trypsin-sensitive surface proteins of bacterial extracellular vesicles are the key inducers of IFN-γ production. Further studies to reveal the specific protein components may be of great value for future immunotherapy.

In conclusion, we present here the first report of using bacterial OMVs as cancer immunotherapeutic agent rather than vaccine or delivery vehicles. The results revealed bacterial extracellular vesicles, especially Gram-negative bacterial OMVs, as a new approach for cancer immunotherapy providing robust therapeutic efficacy without apparent adverse effect. Our mechanism studies showed that theses nano-sized vesicles accumulate in the tumor tissue and produce IFN-γ within the tumor microenvironment to activate antitumor responses. Together, this bacterial extracellular vesicle-based antitumor strategy could bring new insight in the development of novel cancer immunotherapy in the future.

## Methods

**Mice.** BALB/c and C57BL/6 were purchased from the Jackson Laboratory. Male BALB/c background IFN-γ[−/−] mice and C57BL/6 background CXCL10[−/−] mice used. NIHS-Lyst[bg]Foxn1[nu]Btk[xid] mice were purchased from Charles River Laboratory. Mice used in these studies were 6 weeks old in the start of the experiment and were bred and maintained in the specific pathogen-free conditions. All experimental protocols were approved and were performed under the guidance of Institutional Animal Care and Use Committee at Pohang University of Science and Technology, Pohang, Republic of Korea, with the approval number POSTECH-2016-0052-C1.

**Cell culture.** CT26 murine colon adenocarcinoma (American Type Culture Collection, ATCC) and B16BL6 murine melanoma cell lines (ATCC) were grown in minimum essential medium (Gibco) while 4T1 murine mammary carcinoma (ATCC) and MC38 murine colon adenocarcinoma cell lines kindly provided by Dr. Seung-Woo Lee (POSTECH, Pohang, Republic of Korea) were grown in RPMI1640 medium (Gibco). HEK293 cells (ATCC) were grown in Dulbecco's modified Eagle medium (Gibco). All media used for cell cultures were supplemented with 10% fetal bovine serum (Gibco) and 1% Antibiotic-Antimycotic (Invitrogen). Cells were cultured in 37 °C incubator with a humidified atmosphere of 5% $CO_2$.

**Bacterial OMV and extracellular vesicle preparation.** Bacterial OMVs and extracellular vesicles from different Gram-negative and Gram-positive bacteria origins were prepared following the protocol described previously on preparing OMVs derived from E. coli with some modifications[16]. Bacteria cells were cultured overnight on lysogeny broth (1% tryptone, 0.5% yeast extract, 1% NaCl, pH 7.0) at 37 °C with shaking (150 r.p.m.) until the $OD_{600}$ reached 1.5. The cultured cells were pelleted twice at 6,000 g for 20 min at 4 °C. The supernatant was filtered with a filter having 0.45 μm pore size and was concentrated with a QuixStand Benchtop System (Amersham Biosciences) using a 100 kDa hollowfiber membrane (Amersham Biosciences). The concentrate was filtered again using a filter with 0.22 μm pore size and was pelleted by ultracentrifugation at 150,000 g for 3 h at 4 °C. The pellet was suspended in 50% iodixanol and conducted buoyant density gradient of 10%, 40% and 50% iodixanol layers at 200,000 g for 2 h at 4 °C. The fraction containing bacterial OMVs and extracellular vesicles were collected from the third fraction from the top layer. The purified OMVs were filtered with 0.22 μm pore size filter to avoid any bacteria or cell debris contamination. Protein concentration was determined using Bradford assay. The sample was aliquoted and stored at −80 °C until use.

**OMV characterization**. For transmission electron microscope analysis, E. coli wild type and ΔmsbB OMVs were placed on 400-mesh copper grids and stained with 2% uranyl acetate. Images were obtained using a JEM1011 microscope (JEOL) with 100 kV as accelerating voltage. The size of OMVs was measured by dynamic light scattering analyses using Zetasizer 3000HSA (Malvern Instruments) and was analyzed by Dynamic V6 software. Results are from five measurements.

**Antitumor experiment**. For CT26, MC38, B16BL6, and 4T1 syngenic tumor mice models, $2 \times 10^6$ cells were subcutaneously injected to the flank of 6-week-old mice. Male mice were used for all experiments, except for mice injected with 4T1 tumor cells. When the diameter of the tumor reached around 0.8 mm, the mice were blindly allocated to each group for sample treatment. Samples were injected intravenously via tail vein four times with 3 days intervals. Tumor volume (mm$^3$) was calculated as (width)$^2 \times$ (length) $\times 1/2$, using a caliper as described previously[37]. Body weight, body temperature, and tumor volume were measured every 3 days until the end of the experiment. Mice body temperatures were measured using a rectal probe thermometer. Major organs including the lungs, liver, spleen, and kidneys were extracted for histological analyses. Spontaneously metastased 4T1 and B16BL6 colonies in the lungs were counted by naked eye.

**Tissue histological analysis**. Major organs including the lungs, liver, spleen, and kidneys were extracted and fixed in 4% paraformaldehyde for 3 days. The tissues were washed with running water for 20 min and dehydrated with increasing concentration of ethanol and were embedded in paraffin. The paraffin blocks were sectioned to 4 μm thickness and were deparaffinized with xylene and decreasing concentration of ethanol, and were stained with hematoxylin and eosin. The images were obtained using Olympus BX51 microscope (Olympus).

**OMV targeting in vivo**. E. coli ΔmsbB OMVs were labeled with Cy7 mono NHS ester (Amersham Biosciences) by 2 h incubation at 37 °C. Excess Cy7 was removed using ultracentrifugation at 150,000 g for 3 h at 4 °C. Cy7-labeled OMVs (10 μg in total protein) were injected intravenously to normal mice and mice bearing tumor with a diameter of 15 mm (male, 8 weeks old). The increase in the diameter of tumor for targeting experiment is to enhance the signal of OMVs in the tumor tissue to aid better comparison with other organs. Mice were anesthetized and shaved before the intravenous injection of Cy7-labeled OMVs. Cy7 signals were measured using the IVIS spectrum (Caliper Life Sciences), 12 h after the injection. Mice were killed and various tissues were extracted to measure the Cy7 signals in individual organs. After measurement, each organ was weighted for radiant efficiency calculation. Living Image 3.1 software was used for acquiring radiant efficiency and all in vivo imaging experiments were performed at Pohang Center for Evaluation of Biomaterials, Pohang Technopark (Pohang, Republic of Korea).

**Tumor tissue immunohistochemistry**. E. coli ΔmsbB OMVs were injected intravenously to mice bearing tumors. After 12 h, the mice were killed and the tumor tissue was extracted for immunohistochemistry. The tissue was fixed with 4% paraformaldehyde solution for 3 days and was washed in running water for 20 min. The tissue was dehydrated and was embedded in paraffin to make paraffin block. The tissue block was sectioned to 4 μm thickness and was deparaffinized with xylene, and was hydrated with decreasing concentrations of ethanol. The tissue was blocked with 5% horse serum/0.02% Triton X-100 in PBS and was incubated with lab-made anti-E. coli OMV rabbit polyclonal antibodies at room temperature for 2 h[36]. The secondary antibodies conjugated with Alexa Fluor 647 (Molecular Probe, A31573; 1:1,000) were added to the tissue for 1 h. After washing, the cell nucleuses were stained with Hoechst for 10 min at room temperature. For NK cell staining of the tissue, tissues were blocked with 5% horse serum/0.02% Triton X-100 in PBS and were incubated with rat anti-mouse CD49b (BioLegend, 108901; 1:100), which binds to NK cells. Then, biotin-conjugated anti-rat IgM (Bioss Antibodies, bs-0346R-Biotin; 1:500), streptavidin-horseradish peroxidase (R&D, 890803; 1:200), and diaminobenzidine tetrahydrochloride solution (DAB, Biogenex) was added sequentially to form brown spots on the antigen-binding sites. After DAB staining, slides were washed and counterstained with hematoxylin.

For NK, T cells, and IFN-γ staining of the tumor tissue, tumor tissue sections were retrieved and blocked with retrieval and blocking solution (Dako). The tissues were incubated with Alexa fluor 488-conjugated CD49b (BioLegend, 108913; 1:100), biotin-conjugated Armenian hamster anti-mouse CD3ε (Biolegend, 100304; 1:100), and mouse anti-IFN-γ antibody (Bio X Cell, BE0055; 1:500) for 2 h. Then, Alexa Fluor 488-conjugated streptavidin (Invitrogen, S11223; 1:500) and Alexa Fluor 594-conjugated Donkey anti-mouse IgG (Invitrogen, A-21203; 1:500) was added to obtain fluorescent images. Images were visualized using a Olympus confocal microscope (Olympus, FV1000) and were analyzed by FV1000-ASW 3.0 software (Olympus).

**Kinetic quantification of cytokines and chemokines**. E. coli ΔmsbB OMVs were intravenously injected to mice bearing tumors with a diameter of around 15 mm. After the indicated time points, mice were killed and blood serum was collected for serum cytokine measurement. For tumor cytokine and chemokine measurement, tumor tissues were extracted and homogenized. The homogenate was centrifuged

at 1,500 g for 20 min at 4 °C to collect the supernatant. The cytokines and chemokines were measured by enzyme-linked immunosorbent assay following the manufacturer's instructions (R&D Systems).

**Statistical analysis**. For all animal studies, animals of the same gender, age, and genetic background were randomized for grouping. Experiments were not performed in a blinded fashion. Sample sizes were determined on past experiments and previously published results. Data were analyzed by unpaired Student's t-test, one-way or two-way analysis of variance with a Bonferroni post-hoc test using GraphPad Prism 5.0 software. Data were normally distributed and variance between groups was similar. P-values < 0.05 were considered statistically significant. All values are reported as mean ± SD with the indicated sample size. No samples were excluded from analyses.

**Data availability**. The mass spectrometry proteomics data have been deposited to the ProteomeXchange Consortium (http://proteomecentral.proteomexchange.org) via the PRIDE partner repository[43] with the data set identifiers PXD006912 and PXD006913. All other relevant data that support the findings of this study are available from the corresponding authors on reasonable request.

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

## Acknowledgements

This work was supported by the National Research Foundation of Korea (NRF) grant and R&D Convergence Program (CiM) funded by the Korea government (MSIP) (No. 2015R1A2A1A10055961) and NST (National Research Council of Science & Technology) of Republic of Korea (No. CRC-15-02-KRIBB), respectively.

## Author contributions

O.Y.K. and Y.S.G. conceived the idea and designed the research. O.Y.K. prepared OMVs and O.Y.K., and N.T.H.D. carried out OMV characterization experiments. All OMV injections were performed by O.Y.K. and H.T.P.; tumor cell cultures were done by N.T.H.D. H.T.P. and S.J.C. injected the tumors and assisted all animal experiments. J.H.K. helped when performing antitumor experiments on IFN-γ and CXCL10 knockout mice. O.Y.K. carried out immunohistochemistry and kinetics experiments. J.L. performed proteomic analyses of bacterial extracellular vesicles. O.Y.K., N.T.H.D., H.T.P., S.W.L. and Y.S.G. analyzed the data. O.Y.K. and Y.S.G. wrote the manuscript. Other authors revised the manuscript.

## Additional information

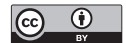

