## [Peer Review File · Nature Communications]

Reviewers' comments:

Reviewer #2 (Remarks to the Author):

This manuscript on outer membrane vesicles (OMVs) as a novel cancer therapy is very interesting and describes how OMVs can suppressive different syngenic tumours in the mouse. Surprisingly, the effect by OMVs can be seen with vesicles isolated from various bacterial species including *E. coli*, *Salmonella enterica*, *Staphylococcus aureus* as well as *Lactobacillus acidophilus*.

1. What components in the OMVs (both Gram-positive as well as Gram-negative) do actually induce the IFN-gamma production?
2. Lipid A –deficient OMVs are isolated from *E. coli*, and the effect is shown to be related to IFN-gamma. Is the same suppressive anti-tumour mechanism responsible for the successful experiments with *Lactobacilli* and *Staphylococci*?
3. How do the OMVs from the Gram-positive species look like in TEM? Are they equally pure?
4. Is there a risk that some component from the culture media plays a role, that is, has the proteome(s) in the OMVs been analysed and contaminants (carried by the OMVs) been excluded?
5. Are OMVs from the Gram-positive bacteria also enriched in the skin tumours?
6. What cell type is producing IFN-gamma?
7. Can we trust the IFN-gamma knock-out mice? Experiments should also be done with anti-IFN-gamma pAbs in order to further verify that this cytokine is the main mechanism of action.
8. To really show the effect and relevance of OMVs in cancer therapy, xenograft tumor models should be included.

Reviewer #3 (Remarks to the Author):

Bacterial outer membrane vesicles suppress tumor by interferon- γ mediated antitumor response

In this study, the authors investigate the potential of bacterial outer membrane vesicles (OMVs) as therapeutic agents to treat cancer via immunotherapy. Bacterial OMVs were generated using genetically modified *E. coli* with inactivated *msbB* to avoid possible adverse effects. While many studies in the past have focused on nanoparticles as vehicles to deliver chemotherapy, this paper is the first to evaluate the potential of bacterial OMVs as immunotherapeutic agents in treating cancer. The OMVs were generated using modified *E. coli* with inactivated *msbB* and demonstrated that the resultant impaired lipid A does not react with TLR4, hypothesizing that this should increase tolerability in vivo. The authors show that while HEK293 cells treated with wildtype OMVs produce high amounts of IL-8, the *msbB*-mutant OMVs elicit no immune response. Further examination confirmed that the OMVs exhibit desirable structure and size. Next, the authors show that administration of OMVs to mice harboring subcutaneous CT26 colon adenocarcinomas results in significant antitumor efficacy. The response was durable and cancer cell-specific and did not produce any noticeable adverse effects.

After demonstrating the antitumor effects of OMVs, the authors confirmed the utility of this approach against multiple tumor models, using OMVs from various bacteria strains. Using fluorescently labeled OMVs, they tracked the distribution of the vesicles in vivo and found that the vesicles accumulate mainly in the tumor tissue of tumor-bearing mice; in contrast, the vesicles accumulated primarily in the spleen and liver of mice lacking tumors. Finally, the authors tried to dissect the mechanism of the OMV-induced antitumor response. They measured cytokine concentrations following injection of OMVs and found that CXCL10 and IFN- γ increased time-dependently in the blood as well as the tumor. Administration of OMVs to CXCL10- or IFN- γ -deficient mice confirmed that the effects of OMVs are IFN- γ dependent, with additional transgenic mice suggesting a role for T cells and NK cells.

The major achievement of this paper is demonstrating for the first time that bacterial OMVs are capable of effectively inducing long-term antitumor immune responses. Additionally, the authors reveal that this effect was IFN- γ -dependent and applicable to multiple tumor models. The findings are interesting and novel in the field of cancer immunology. The authors generally provide adequate evidence to support their claims, and the experiments include appropriate controls. The inclusion of several tumor models increases confidence in this proof-of-concept study that the described effect of OMVs is reproducible and applicable to a variety of cancer types – albeit with varying efficacy, depending on the aggressiveness of the model. The work described may be translatable to the clinic, though several questions remain to be explored. The work expands on previous literature that uses nanoparticles for tumor therapy, and the authors treated the literature fairly. Details of the methodology are mostly sufficient to allow the experiments to be reproduced, though an expanded description of the methods for production of OMVs would be desirable. Standardized scientific nomenclature and abbreviations are used. The abstract, introduction, and conclusions are all appropriate, though the discussion would benefit from inclusion of additional substance.

The manuscript would, nonetheless, benefit from some revisions.

1) For Figure 1d (as well as Figures 2a and 2d), is a longer time course available? Not only would the tumor volume be interesting – as Figure 2e suggests that a rebound is possible – but also survival data would be much more compelling.

2) In Figure 1e, it is mentioned that distinct phenotypical and histological changes were observed, but the changes are not specified. A more detailed examination should be added.

3) For Figure 1g, it is not clear if the mice were injected in the middle of the two flanks (as stated in the text) or in the top flank (as stated in the figure legend) for the tertiary challenge. The description should be uniform.

4) For Figures 2b and 2c, an explanation of how the lung metastasis were counted could be provided. Also a representative image should be included.

5) Regarding Figure 3b, it is mentioned that the OMVs accumulate mainly in the spleen and liver in mice lacking tumors while they are found primarily in the tumor in tumor-bearing mice. Although the EPR effect is offered as an explanation, it is still striking that there are hardly any OMVs found in the liver or spleen of tumor-bearing mice, particularly in the latter, which is a secondary lymphoid organ in addition to being a filtration organ. A possible explanation should be provided. An enrichment of signal in the tumor would be reasonable, but exclusive accumulation in the tumor is highly surprising, particularly because the total signal is so much lower than in the no tumor control. Where did all of the other OMVs go? Importantly, the tumor targeting experiment was performed with OMVs having a diameter of 15 μm , whereas the efficacy studies were performed with OMVs having a diameter of 0.8 μm . An explanation for this deviation should be provided.

Also, there appears to be splenomegaly for the spleen isolated from a tumor-bearing mouse treated with OMVs relative to the non-tumor-bearing control. It would be interesting to see whether this was observed in a tumor-bearing mouse that was not treated with OMVs. Indeed, the histology of the spleen isolated from a mouse treated with OMVs looks inflamed relative to the control spleen (Fig. S3c). Questions over safety are similarly raised by the loss of body weight after injection of OMVs (Fig. S3b).

6) For Figure 3c, it should be explained how the radiant efficiency was calculated. Also, the spleen appears to yield the highest signal, but the picture shown in Figure 3b suggests that the majority of the dose accumulates in the liver. Perhaps the labels in the graph were switched. Finally, the lung and kidney appear to yield signals that are at least half of that yielded by the liver, yet there is absolutely no signal emanating from these organs in Figure 3b.

7) In Figure 3d, the image showing the staining/contrast for OMVs should be enhanced so the stained OMVs are more visible.

8) For Figures 4a and 4b, why are the cytokines observed in the serum before they are detected in the tumor itself? Wouldn't one expect a Th1 response to originate in the tumor (and not much sooner than 24 hours, as consistent with Figure 4b)? What is the origin of the early response in the blood? This group has previously reported (J Immunol, 2013) that a Th17 response is observed in response to the bacterially derived product, as expected. The Th1 response that they observed in that study was related to antigen specificity upon challenge with bacteria. It is not apparent why there would be an antigen-specific response to the tumor following administration of the OMVs, which do not have shared antigens. The OMVs should not be particularly stimulatory to the innate immune system, as they do not stimulate TLR4, which would be the anticipated means of activating the host immune system. What is stimulating the immune system if the bacterial endotoxin function has been removed (Fig. S1)?

Moreover, the group also reported (Small, 2015) that the inflammatory effects of OMVs resolve by 24 hours, which differs from what is observed herein. This may be a result of intraperitoneal injection versus intravenous injection, but it is unlikely that the latter would clear before the former; supposedly the difference is owing to the presence of a tumor, but, again, it is not clear why. Finally, why is IL-12p40 (homodimer of p40) detected at elevated levels, while IL-12p70 (heterodimer containing p35 and p40) is not (Fig. S5)? The latter is the active form of IL-12.

9) In Figure 4d, the line and "n.s." written above the OMV data should be removed, as the asterisks suggest that the data are significant. This was likely accidentally copied and pasted.

10) In Figure 5c, why is there necrotic tissue surrounding the NK cells if these cells are purported to be dysfunctional in this transgenic model? These data seem to go against the claim provided.

11) In the Discussion, it is mentioned that bacterial extracellular vesicles are present in the blood and elsewhere in the body; why are these not effective at promoting antitumor immunity? Why are OMVs required? Is it a matter of dose? Also, it is claimed that the mechanisms studies show that OMVs "specifically target and activate immune cells to produce IFN-g within the tumor microenvironment," but this is not shown. What evidence is there that the vesicles specifically target and activate immune cells?

The statistics in the figures are sometimes confusing, as it is not always clear which results are compared to which; this should be fixed. In the methods section, it is mentioned that body temperature was measured; it should be explained how that was done. The NK staining in Figure 5c should be mentioned in the methods section as well as the measurement of IL-8 cytokine from Supplementary Figure 1.

The following minor wording revisions are suggested (additions are bolded):

- 1) "However, nano-sized particles can easily flow through the blood and lymphatic vessels and can readily interact with or be ingested by immune cells, giving them great potential as immunostimulatory agents."
- 2) "This antitumor response of bacterial OMVs was durable, and secondary and tertiary re-challenges of tumor were fully rejected by mice that were cured from primary challenge."
- 3) "we used Gram-negative bacterial OMVs derived from genetically modified *Escherichia coli*, whose gene encoding lipid A acyltransferase (*msbB*), the lipid component of lipopolysaccharide, had been inactivated (*E. coli msbB*^{-/-}, Δ *msbB*)."
- 4) "Furthermore, all mice injected with 1×10^9 CFU died within 48 h after the injection and most of the mice developed systemic inflammatory response syndrome symptoms like the formation of eye exudates or piloerection hypothermia."

In summary, this reviewer believes that this manuscript is suitable for publication in Nature Communications following major revisions. The work is of importance to researchers in the field, though the methodology could be more rigorous to enhance support for the stated conclusions. The efficacy data in Figure 1 are extremely provocative, but the lack of mechanism – particularly relating to induction of antitumor immunity but also for tumor targeting – should be addressed.

Response to Reviewers' Comments

Reviewer #2

This manuscript on outer membrane vesicles (OMVs) as a novel cancer therapy is very interesting and describes how OMVs can suppressive different syngenic tumours in the mouse. Surprisingly, the effect by OMVs can be seen with vesicles isolated from various bacterial species including *E. coli*, *Salmonella enterica*, *Staphylococcus aureus* as well as *Lactobacillus acidophilus*.

1. What components in the OMVs (both Gram-positive as well as Gram-negative) do actually induce the IFN-gamma production?
 - To investigate which components in the bacterial extracellular vesicles actually induce IFN-gamma production, we have performed additional experiments using heat- and trypsin-treated *E. coli* $\Delta msbB$ OMVs and *S. aureus* wildtype extracellular vesicles. Both heat- and trypsin-treated *E. coli* $\Delta msbB$ OMVs and *S. aureus* wildtype extracellular vesicles did not induce any IFN- γ production, suggesting that the trypsin-sensitive surface vesicular proteins are the key factors involved in IFN- γ production.
 - Furthermore, we identified 200 and 476 vesicular proteins by the proteomic analysis of *E. coli* $\Delta msbB$ OMVs and *S. aureus* wildtype extracellular vesicles, respectively. Further analysis showed that neither yeast nor cow proteins were identified from proteomic analyses, suggesting that purified *E. coli* $\Delta msbB$ OMVs and *S. aureus* wildtype extracellular vesicles are free of potential contaminants from the culture media.
 - Taken together, these results suggest that trypsin-sensitive surface proteins of bacterial extracellular vesicles are the key inducers of IFN- γ production. Further studies to reveal the specific protein components may be of great value for future immunotherapy.
 - We added this data as NEW Supplementary Figure 15, Supplementary Figure 16, Supplementary Table 1, and Supplementary Table 2 and added this information

on the discussion and methods sections on page 10 (lines 1-22), page 11 (lines 1,2) and on Supplementary methods, respectively.

2. Lipid A –deficient OMVs are isolated from *E. coli*, and the effect is shown to be related to IFN-gamma. Is the same suppressive anti-tumour mechanism responsible for the successful experiments with Lactobacilli and Staphylococci?

- To find out if the same suppressive anti-tumor mechanism of IFN-gamma playing the key role is responsible for both Gram-negative- and Gram-positive-derived extracellular vesicles, we performed the same anti-tumor experiments using *S. aureus* and *L. acidophilus* extracellular vesicles on IFN-gamma knockout mice. When Gram-positive *S. aureus* and *L. acidophilus* extracellular vesicles were treated to IFN- γ deficient mice, antitumor response was not observed. These observations imply that IFN- γ plays an important role in inducing both Gram-negative and Gram-positive bacterial extracellular vesicle-induced antitumor response.
- We have added these results as NEW Supplementary Figure 12 and this information on page 8 (lines 7-11).

3. How do the OMVs from the Gram-positive species look like in TEM? Are they equally pure?

- As suggested, we took TEM images of *S. aureus* and *L. acidophilus* extracellular vesicles. Both extracellular vesicles have vesicular structures of around 25 nm in diameters and were equally pure as Gram-negative *E. coli* OMVs.
- We have added the TEM images of *S. aureus* and *L. acidophilus* extracellular vesicles as NEW Supplementary Figure 5 and on page 6 (line 10).

4. Is there a risk that some component from the culture media plays a role, that is, has the proteome(s) in the OMVs been analysed and contaminants (carried by the OMVs) been excluded?

- As suggested, we performed the proteomic analysis of *E. coli* $\Delta msbB$ OMVs and *S. aureus* wildtype extracellular vesicles and identified 200 and 476 *E. coli* $\Delta msbB$ and *S. aureus* vesicular proteins, respectively.
- We cultured *E. coli* $\Delta msbB$ and *S. aureus* in Luria-Bertani broth which are compo

sed of NaCl, yeast extracts, and tryptone (peptides formed by the digestion of trypsin of cow casein). Further analysis showed that neither yeast nor cow proteins were identified from proteomic analyses.

- Taken together, the purified *E. coli* Δ *msbB* OMVs and *S. aureus* wildtype extracellular vesicles used in this study are free of potential contaminants from the culture media.
- We added this data as NEW Supplementary Figure 16, Supplementary Table 1, and Supplementary Table 2 and added this information on the discussion on page 10 (lines 14-22), page 11 (lines 1,2), and Supplementary methods sections.

5. Are OMVs from the Gram-positive bacteria also enriched in the skin tumours?

- As suggested, we carried out *in vivo* targeting experiment using *S. aureus*-derived extracellular vesicles. When we systemically administered *S. aureus* extracellular vesicles labeled with Cy7 to mice bearing tumors, *S. aureus* extracellular vesicles were accumulated in the tumor tissue suggesting that Gram-positive bacterial extracellular vesicle also target to the skin tumors though the organ distribution pattern is somewhat different from that of *E. coli* OMVs.
- We have added this result as NEW Supplementary Figure 8 on page 7 (lines 5-7).

6. What cell type is producing IFN-gamma?

- To answer the question “what cell type is producing IFN-gamma”, we carried out further immunofluorescence study using OMV-treated tumor tissues. We found that IFN-gamma was co-localized with NK and T cells in the tumor tissues of mice 48 h after the OMVs injection suggesting that both NK and T cells produce IFN-gamma after OMV injection (NEW Figure 5a and NEW Supplementary Figure 13).
- These results are consistent with the previously known fact as mentioned in the manuscript: the major producers of IFN-gamma are NK and T cells [*Adv Immunol* **2007**, 96:41-101]. Moreover, in line with these results, we also observed that NK cells accumulate in the tumor necrotic area after OMV injection (Figure 5b) and that OMV-induced antitumor response is not observed in NIHS-*Lyst*^{bg}*Foxn1*^{nu}*Btk*^{xid} mice deficient with the major producers of IFN- γ , the NK and T cells (Figure 5c and d).

Furthermore, OMV-treatment to athymic nude (Nu/J (Fox1nu/Fox1nu), F120) mice with T cell deficiency, showed about 50% of OMV antitumor effect (NEW Supplementary Figure 14).

- Taken together with IFN-gamma knockout mice (Figure 4d and NEW Supplementary Figure 12) and anti-IFN-gamma antibody studies (NEW Supplementary Figure 11), our results provide evidence that the NK and T cells are the major producers of IFN-gamma and that IFN-gamma plays important roles in mediating OMV-induced antitumor responses.
 - We have added these results as NEW Figure 5a and NEW Supplementary Figure 13) on page 8 (lines 12-14).
7. Can we trust the IFN-gamma knock-out mice? Experiments should also be done with anti-IFN-gamma pAbs in order to further verify that this cytokine is the main mechanism of action.
- As the reviewer suggested, we have performed additional neutralizing experiment with anti-IFN-gamma antibodies to further verify that this cytokine is the main mechanism of OMV-induced antitumor response. Mice injected with mouse monoclonal anti-IFN- γ antibody prior to OMV treatment did not show tumor regression while mice injected with isotype IgG₁ antibody prior to OMV treatment showed complete regression of tumor.
 - We have added this information as NEW supplementary Figure 11 on page 8 (lines 3-7) and on Supplementary methods section.
8. To really show the effect and relevance of OMVs in cancer therapy, xenograft tumor models should be included.
- We agree with the reviewer that to show the relevance of OMVs as potential cancer therapeutic agent, we should show the antitumor effect of OMVs using xenograft tumor models.
 - However, in order for us to make xenograft tumor models in mice, we had to use athymic nude mice with T cell deficiency or NIHS-Lyst^{bg}Foxn1^{nu}Btk^{xid} mice with both NK and T cell deficiency. However, as we claimed in our manuscripts, this OMV antitumor effect primarily requires IFN-gamma production by NK and T

cells and probably other immune cells for full tumor regression. In fact, OMV-induced antitumor response was not observed in NIHS-Lyst^{bg}Foxn1^{nu}Btk^{xid} mice deficient with the major producers of IFN- γ , the NK and T cells (Figure 5c and d). Furthermore, OMV-treatment to athymic nude (Nu/J (Fox1nu/Fox1nu), F120) mice with T cell deficiency, showed about 50% of OMV antitumor effect (NEW Supplementary Figure 14). Taken together, we can speculate that the antitumor effect of OMVs could not be fully observed in the xenograft tumor model using immuno-deficient mice such as athymic nude mice or NIHS-Lyst^{bg}Foxn1^{nu}Btk^{xid} mice.

Reviewer #3

Bacterial outer membrane vesicles suppress tumor by interferon- γ mediated antitumor response.

In this study, the authors investigate the potential of bacterial outer membrane vesicles (OMVs) as therapeutic agents to treat cancer via immunotherapy. Bacterial OMVs were generated using genetically modified *E. coli* with inactivated *msbB* to avoid possible adverse effects. While many studies in the past have focused on nanoparticles as vehicles to deliver chemotherapy, this paper is the first to evaluate the potential of bacterial OMVs as immunotherapeutic agents in treating cancer. The OMVs were generated using modified *E. coli* with inactivated *msbB* and demonstrated that the resultant impaired lipid A does not react with TLR4, hypothesizing that this should increase tolerability *in vivo*. The authors show that while HEK293 cells treated with wildtype OMVs produce high amounts of IL-8, the *msbB*-mutant OMVs elicit no immune response. Further examination confirmed that the OMVs exhibit desirable structure and size. Next, the authors show that administration of OMVs to mice harboring subcutaneous CT26 colon adenocarcinomas results in significant antitumor efficacy. The response was durable and cancer cell-specific and did not produce any noticeable adverse effects.

After demonstrating the antitumor effects of OMVs, the authors confirmed the utility of this approach against multiple tumor models, using OMVs from various bacteria strains. Using fluorescently labeled OMVs, they tracked the distribution of the vesicles *in vivo* and found that the vesicles accumulate mainly in the tumor tissue of tumor-bearing mice; in contrast, the

vesicles accumulated primarily in the spleen and liver of mice lacking tumors. Finally, the authors tried to dissect the mechanism of the OMV-induced antitumor response. They measured cytokine concentrations following injection of OMVs and found that CXCL10 and IFN- γ increased time-dependently in the blood as well as the tumor. Administration of OMVs to CXCL10- or IFN- γ -deficient mice confirmed that the effects of OMVs are IFN- γ dependent, with additional transgenic mice suggesting a role for T cells and NK cells.

The major achievement of this paper is demonstrating for the first time that bacterial OMVs are capable of effectively inducing long-term antitumor immune responses. Additionally, the authors reveal that this effect was IFN- γ -dependent and applicable to multiple tumor models. The findings are interesting and novel in the field of cancer immunology. The authors generally provide adequate evidence to support their claims, and the experiments include appropriate controls. The inclusion of several tumor models increases confidence in this proof-of-concept study that the described effect of OMVs is reproducible and applicable to a variety of cancer types – albeit with varying efficacy, depending on the aggressiveness of the model. The work described may be translatable to the clinic, though several questions remain to be explored. The work expands on previous literature that uses nanoparticles for tumor therapy, and the authors treated the literature fairly. Details of the methodology are mostly sufficient to allow the experiments to be reproduced, though an expanded description of the methods for production of OMVs would be desirable. Standardized scientific nomenclature and abbreviations are used. The abstract, introduction, and conclusions are all appropriate, though the discussion would benefit from inclusion of additional substance.

- We thank the reviewer for the clarification of our manuscript and the positive comments.
- As suggested, we added expanded description of the methods for production of OMVs and extracellular vesicles in the Methods section.
- As suggested, we added additional discussion points in the Discussion section.

The manuscript would, nonetheless, benefit from some revisions.

1. For Figure 1d (as well as Figures 2a and 2d), is a longer time course available? Not only would the tumor volume be interesting – as Figure 2e suggests that a rebound is

possible – but also survival data would be much more compelling.

- We fully agree with a reviewer that it would be much more compelling if we show the survival data together with tumor volume. However, due to the ethical issue, as stated in our institute animal experiment ethics, we sacrificed the control mice injected with PBS having tumors of certain size. In addition, we could not show the survival data, as our animal experiment approved for this project did not allow survival test involving severe pain. However, for mice treated with OMVs having complete regression of tumors, we did not observe any rebound of tumor even after long periods of more than 5 weeks.
 - Considering the reviewer's comment regarding tumor rebound in Figure 2e, one of the mice (total n=5 per each group) in the Gram-positive extracellular vesicle-treated each group did not show complete regression of tumor during the first week of injection: this tumor subsequently grew back to certain size at the end of the experiment as shown in the Figure 2e. Since we did not exclude any data for our animal experiments as mentioned in Methods, our graphs in Figure 2e were shown as if there is tumor volume rebound after extracellular vesicle-treatment. Thus, we have performed another set of experiment using *S. aureus* wildtype extracellular vesicle-treated group. As a result, we observed complete regression of tumors for all mice. Furthermore, we have monitored the mice until 65 days after the tumor challenge but did not observe any tumor rebound.
 - We have added the results as NEW Supplementary Figure 6 on page 6 (lines 13-17).
2. In Figure 1e, it is mentioned that distinct phenotypical and histological changes were observed, but the changes are not specified. A more detailed examination should be added.
 - As suggested by the reviewer, we have added more detailed explanation on what kind of changes were observed on page 5 (lines 6,7).
 3. For Figure 1g, it is not clear if the mice were injected in the middle of the two flanks (as stated in the text) or in the top flank (as stated in the figure legend) for the tertiary challenge. The description should be uniform.

- We thank the reviewer for the correction. The tumor cells for the tertiary challenge were injected in the middle of the two flanks as state in the text. We have corrected the figure legend as such.
4. For Figures 2b and 2c, an explanation of how the lung metastasis were counted could be provided. Also a representative image should be included.
 - As suggested by the reviewer, we added the explanation on how the lung metastasis were counted in the materials and methods section on page 13 (lines 13,14). In addition, as suggested, we have included representative lung images to each figure.
 5. Regarding Figure 3b, it is mentioned that the OMVs accumulate mainly in the spleen and liver in mice lacking tumors while they are found primarily in the tumor in tumor-bearing mice. Although the EPR effect is offered as an explanation, it is still striking that there are hardly any OMVs found in the liver or spleen of tumor-bearing mice, particularly in the latter, which is a secondary lymphoid organ in addition to being a filtration organ. A possible explanation should be provided. An enrichment of signal in the tumor would be reasonable, but exclusive accumulation in the tumor is highly surprising, particularly because the total signal is so much lower than in the no tumor control. Where did all of the other OMVs go?
 - We fully agree with the reviewer's concern that it is strange that tumor tissue alone has highest signal for Cy7 while the spleen has almost no Cy7 signals in tumor-bearing mice. However, the technique in the *in vivo* imaging system uses to calculate the RELATIVE radiation efficiency has many limitations as the value can change according to various factors. For example, the value we obtain as radiation efficiency depends on various factors like depth of the organ and the angle of the detection camera to each organ [*J Photochem Photobiol B* **2010**, 98:77-94]. The *in vivo* imaging system measures the RELATIVE radiation efficiency of the image in one field. In Figure 3b, the signal from tumor is too high in tumor bearing mice making the relative Cy7 signals for other organs, especially in liver and spleen as the reviewer mentioned, not detectable. Therefore, we should not expect to get precise values of the Cy7 signals for each organ for all individual images, but we should compare the relative radiation signal on that particular acquired image. Thus, we have changed the exposure

intensity so that the signals for other organs in tumor-bearing mice are shown to some degree. We replaced Figure 3b with new Figure 3b and re-calculated the radiation efficiency for each organ for Figure 3c.

Importantly, the tumor targeting experiment was performed with OMVs having a diameter of 15 μm , whereas the efficacy studies were performed with OMVs having a diameter of 0.8 μm . An explanation for this deviation should be provided.

- The aim for the tumor targeting experiment was to show the accumulation of the vesicles in the tumor tissue by comparing with the other major organs. However, if we did the experiment on mice having tumor of 0.8 mm in diameter, this would be too small for comparison. Thus, we used mice having tumors around 15 mm in diameter for IVIS experiment.
- We added this information on the manuscript on page 14 (lines 3,4).

Also, there appears to be splenomegaly for the spleen isolated from a tumor-bearing mouse treated with OMVs relative to the non-tumor-bearing control. It would be interesting to see whether this was observed in a tumor-bearing mouse that was not treated with OMVs. Indeed, the histology of the spleen isolated from a mouse treated with OMVs looks inflamed relative to the control spleen (Fig. S3c). Questions over safety are similarly raised by the loss of body weight after injection of OMVs (Fig. S3b).

- As the reviewer is concerned, it is well-known that enlargement of spleen, the splenomegaly, is present in tumor-bearing mice as the evidence of immunological activity against the tumor [*Nature* **1965**, 205:918-919; *Biomed Environ Sci* **2014**, 1:17-26]. As the reviewer pointed out, we have also observed splenomegaly on tumor bearing mice with or without OMV treatment compared to normal mice. In addition, when we compared the weights of the spleens extracted from tumor bearing mice with or without OMV treatment, there was no statistically significant difference. Therefore, we could speculate that the splenomegaly is due to the presence of tumor but not OMV injections.
- For the histology image of spleen in Supplementary Figure 3c, we have replaced the image with images obtained from other spleen tissue image of the same mice group to avoid any confusion.

- Lastly, as the reviewer pointed out, there was slight decrease in the body weight after the first two OMV injections in Supplementary Figure 3b. We think that this change in the body weight might be because of the strong anti-tumor response following the regression of tumor tissue but not because of the toxicity of the OMVs since we could not observe any phenotypical change or mice behavior compared to control. In addition, when we monitor the body weight for long periods after OMVs treatment, we observed that the body weight of the OMV-treated group increases normally as the mice matures whereas the PBS control group's body weight decreases significantly due to enlarged tumors. However, in-depth safety evaluation of the OMV injections should be evaluated in the future to validate their safety for clinical use.
 - We have added the new spleen image on revised Supplementary Figure 3c and added more information on in-depth safety evaluation issue for clinical use (page 5, lines 3-5).
6. For Figure 3c, it should be explained how the radiant efficiency was calculated. Also, the spleen appears to yield the highest signal, but the picture shown in Figure 3b suggests that the majority of the dose accumulates in the liver. Perhaps the labels in the graph were switched. Finally, the lung and kidney appear to yield signals that are at least half of that yielded by the liver, yet there is absolutely no signal emanating from these organs in Figure 3b.
- Here, we think there was misunderstanding regarding Figure 3c. As the reviewer is concerned, it is true that the Cy7 fluorescence intensity of OMVs was the strongest in the liver for whole organ image (Figure 3b) for normal mice. In Figure 3c, however, we have divided the fluorescence intensity of Cy7 of each organ by their weight to normalize fluorescence intensity by each organ weight. Therefore, the spleen had the highest signal in Figure 3c.
 - We have informed in our manuscript that the radiation efficiencies were divided by each organ weights on the results, figure legends and methods sections, on page 7 (lines 3-5), page 14 (lines 8,9), and page 24 (line 6), respectively. Considering the reviewer's comment, we agree that showing only the graph in Figure 3c might cause confusion to the readers. Therefore, we have added a NEW Supplementary Figure 7 without normalizing the fluorescence intensity

by the tissue weight on page 7 (lines 1,2).

7. In Figure 3d, the image showing the staining/contrast for OMVs should be enhanced so the stained OMVs are more visible.

- We have enhanced the contrast, as the reviewer suggested, and further enlarged the images to make OMVs more visible in Figure 3d.

8. For Figures 4a and 4b, why are the cytokines observed in the serum before they are detected in the tumor itself? Wouldn't one expect a Th1 response to originate in the tumor (and not much sooner than 24 hours, as consistent with Figure 4b)? What is the origin of the early response in the blood? This group has previously reported (J Immunol, 2013) that a Th17 response is observed in response to the bacterially derived product, as expected. The Th1 response that they observed in that study was related to antigen specificity upon challenge with bacteria. It is not apparent why there would be an antigen-specific response to the tumor following administration of the OMVs, which do not have shared antigens. The OMVs should not be particularly stimulatory to the innate immune system, as they do not stimulate TLR4, which would be the anticipated means of activating the host immune system. What is stimulating the immune system if the bacterial endotoxin function has been removed (Fig.S1)?

- Since we have injected OMVs through tail-vein systemic infection, OMVs, right after entering the body through the blood vessel, circulate the body and are brought to certain organs or tissues afterwards. Therefore, it is likely that blood circulating monocytes, neutrophils, NK cells, and other immune cells are the first line of immune cells the OMVs first encounter. As these blood-born immune cells get activated by OMVs in the blood, they would produce cytokines faster in the blood serum than immune cells in the tumor tissue.
- Our prior aim in the previous study published in the Journal of Immunology in 2013, was to induce antigen-specific immune response to observe OMV-induced vaccination effect (prevention through memory response). This is quite different from our current study of applying OMVs as tumor immunotherapeutic agents. Both are similar in that OMVs are used as inducers of immune response. However, for prevention of inflammatory response, induction of antigen-specific

memory response is very important as the whole concept of vaccination is about preventing the specific pathogens having the vaccinated antigen. For instance, we wanted to make sure that the OMV-induced inflammatory response is only specific to fighting OMVs and bacteria, and do not cause random toxic response systemically. On the other hand, our goal in this study is to apply the ability of OMVs to induce immune response in treating cancer by shifting the immune surveillance system in the tumor microenvironment to attack tumor cells. In this process, memory response should be activated both against OMVs and tumor cells. Thus, 2nd and 3rd challenge of the tumor challenge was eliminated after OMV treatment. However, detailed mechanism should be performed in the future.

- In response to the reviewer's question regarding what is stimulating the immune system when endotoxin function is removed, we would like to emphasize two things. First, it is true that Gram-negative endotoxin LPS is recognized by the TLR4 that in turns activates the innate immune system. However, there are many other pattern recognition receptors including other TLRs, NLRs or CDS that are responsible for recognizing different pathogenic molecules. For example, bacterial peptidoglycan is recognized by both TLR2 and NOD1 or NALP1/3. Second, in case of bacterial extracellular vesicles in our studies, we used different extracellular vesicles derived from various bacterial strains including Gram-positive bacteria which do not express Gram-negative endotoxin LPS. In addition, to examine which component of the bacterial extracellular vesicles are important inducer of IFN-gamma production, we carried out additional experiment using heat- and trypsin-treated *E. coli* Δ *msbB* OMVs and *S. aureus* extracellular vesicles and found that trypsin-sensitive surface vesicular proteins were responsible (NEW Supplementary Figure 15). Thus, we could assume that in our study, the immune system is activated by different components other than LPS and that trypsin-sensitive surface vesicular proteins are responsible for IFN-gamma production in both Gram-negative and Gram-positive extracellular vesicles. It would be of great value if we find out these specific components and the recognition receptors involved in future studies.

Moreover, the group also reported (Small, 2015) that the inflammatory effects of OMVs resolve by 24 hours, which differs from what is observed herein. This may be a result of intraperitoneal injection versus intravenous injection, but it is unlikely that

the latter would clear before the former; supposedly the difference is owing to the presence of a tumor, but, again, it is not clear why. Finally, why is IL-12p40 (homodimer of p40) detected at elevated levels, while IL-12p70 (heterodimer containing p35 and p40) is not (Fig. S5)? The latter is the active form of IL-12. Finally, why is IL-12p40 (homodimer of p40) detected at elevated levels, while IL-12p70 (heterodimer containing p35 and p40) is not (Fig. S5)? The latter is the active form of IL-12.

- As shown in the study published in Small 2015, OMVs were still present in the liver, kidney and spleen even after 24 h whereas only small amount of OMVs were detected in the blood serum. This suggests that OMVs when injected, circulates around the body through the blood and are brought to special organs for immune response activation or excretion within 24 h of injection. However, when there is tumor, it is likely that some OMVs end up in the tumor tissue through enhanced permeability and retention, the EPR, effect. This may be why our *in vivo* imaging system data in Figure 3 showed OMVs in tumor tissue as well as in other tissues like the liver and spleen 12 h after the injection. In addition, although the reviewer commented that in the Small article, inflammatory effect of OMVs were resolved by 24 h, according to the article, systemic inflammatory response as shown by ICAM-1 levels in the bronchoalveolar lavage fluid was the highest at 24 h. This suggests that after peritoneal injection of the OMVs, inflammatory response in the distant organs is still elevating after 24 h. Thus, although we cannot argue that the mode of action for the OMVs in this study and our previous study published in Small is exactly the same as the injection site, dosage and injection number differs between the two studies, we could say that there are some co-relations to how the OMVs induce immune response in distant organs, tissues, or tumors at later time points.
- It is true that IL-12 is heterodimer composed of IL-12p35 and IL-p40 subunits. However, it is known that IL-12p40 acts as an antagonist of IL-12p70 by competitive binding to the receptor and that IL-12p40 alone can induce the activation of IL-23. Importantly, most of the antigen presenting cells only produce IL-12p40 and can not produce IL-12p35. Thus, we could observe more IL-12p40 production after OMV administration since most of the immune cells involved, such as the macrophages and dendritic cells only produce IL-12p40.

9. In Figure 4d, the line and “n.s.” written above the OMV data should be removed, as the asterisks suggest that the data are significant. This was likely accidentally copied and pasted.

- We thank the reviewer for the correction. We have corrected Figure 4d.

10. In Figure 5c, why is there necrotic tissue surrounding the NK cells if these cells are purported to be dysfunctional in this transgenic model? These data seem to go against the claim provided.

- We think there was some misunderstanding regarding the strain of the mice used for Figure 5c. The tumor tissue sections shown in Figure 5c are derived from WILDTYPE MICE with normal NK cell function but not from NIHS-Lyst^{bg}Foxn1^{nu}Btk^{xid} MICE. This confusion was due to our mistake of not providing the information regarding the mice strain in the manuscript. We have revised our manuscript and added information about mouse strain on page 8 (lines 14, 15) and rewrote the Figure Legend of Figure 5 on page 23.

11. In the Discussion, it is mentioned that bacterial extracellular vesicles are present in the blood and elsewhere in the body; why are these not effective at promoting antitumor immunity? Why are OMVs required? Is it a matter of dose? Also, it is claimed that the mechanisms studies show that OMVs “specifically target and activate immune cells to produce IFN-g within the tumor microenvironment,” but this is not shown. What evidence is there that the vesicles specifically target and activate immune cells?

- All Gram-negative and Gram-positive bacteria produce OMVs and extracellular vesicles, respectively, to their surroundings and therefore this is also true for bacteria composing our body’s microflora. Human body has various bacteria cells and in places like the gut, there are more bacteria cells than human cells. Thus, bacteria-derived extracellular vesicles are naturally found within our body’s fluids as well as in the feces. However, these bacteria-derived extracellular vesicles are part of our symbiotic system and are seemed to be considered self and our body’s immune system does not respond to them until they are found in places where they should not be found or in large amounts. In our study, we injected bacterial extracellular vesicles through tail-vein systemic injection. As the reviewer

mentioned, the amount of bacterial extracellular vesicles injected is considered very high in terms of their biological concentration. Thus, we speculate that our systemically injected bacterial extracellular vesicles were first brought to the tumor tissue *via* enhanced permeability and retention (EPR) effect which triggered antitumor effect on tumors as the immune cells in the tumor tissue considered them foreign or not normal. Nevertheless, future in-depth studies should be performed to validate this hypothesis.

- We agree with the reviewer that we have not directly shown that bacterial extracellular vesicles specifically target and activate immune cells. Thus, we have revised the sentence to bacterial extracellular vesicles accumulate in the tumor tissue and produce IFN-gamma within the tumor microenvironment to activate antitumor responses on page 11 (lines 7,8).

The statistics in the figures are sometimes confusing, as it is not always clear which results are compared to which; this should be fixed. In the methods section, it is mentioned that body temperature was measured; it should be explained how that was done. The NK staining in Figure 5c should be mentioned in the methods section as well as the measurement of IL-8 cytokine from Supplementary Figure 1.

- We went through all the figures and made sure that we indicate which group was compared to which for all figures with statistics information on the figure legends.
- As suggested, we have added the protocols for mice body temperature measurement, NK staining, and IL-8 cytokine measurements on the Methods and Supplementary methods section.

The following minor wording revisions are suggested (additions are bolded):

1. “However, nano-sized particles can easily flow through the blood and lymphatic vessels and can readily interact with or be ingested by immune cells, giving them great potential as immunostimulatory agents.”
 - We thank the reviewer for the correction. We have revised the manuscript as suggested on page 2 (line 4).

2. “This antitumor response of bacterial OMVs was durable, and secondary and tertiary re-challenges of tumor were fully rejected by mice that were cured from primary challenge.”
 - We thank the reviewer for the correction. We have revised the manuscript as suggested on page 3 (lines 2,3).
3. “we used Gram-negative bacterial OMVs derived from genetically modified *Escherichia coli*, whose gene encoding lipid A acyltransferase (*msbB*), the lipid component of lipopolysaccharide, had been inactivated (*E. coli msbB*^{-/-}, Δ *msbB*).”
 - We thank the reviewer for the correction. We have revised the manuscript as suggested on page 3 (lines 14-15).
4. “Furthermore, all mice injected with 1×10^9 CFU died within 48 h after the injection and most of the mice developed systemic inflammatory response syndrome symptoms like the formation of eye exudates or piloerection hypothermia.”
 - We thank the reviewer for the correction. We have revised the manuscript as suggested on page 4 (line 21).

In summary, this reviewer believes that this manuscript is suitable for publication in Nature Communications following major revisions. The work is of importance to researchers in the field, though the methodology could be more rigorous to enhance support for the stated conclusions. The efficacy data in Figure 1 are extremely provocative, but the lack of mechanism – particularly relating to induction of antitumor immunity but also for tumor targeting – should be addressed.

REVIEWERS' COMMENTS:

Reviewer #2 (Remarks to the Author):

I have no further comments and congratulate the authors to an excellent study.

Reviewer #3 (Remarks to the Author):

Reviewer 3 feels that the authors did a commendable job of addressing the concerns articulated and believes that the manuscript is now suitable for publication in Nature Communications. Some matters still remain unresolved (e.g., the mechanism underlying the induction of innate immune activation [presumably mediated by peptidoglycans, based on the trypsin data, though the specific pattern recognition receptor remains undefined] -- which is likely the most interesting aspect of this project); still, the conclusions are generally well supported by the data presented, and the findings appear to be quite robust.

Response to Reviewers' Comments

Reviewer #2

I have no further comments and congratulate the authors to an excellent study.

- We thank the reviewer for the positive comment regarding our revised manuscript.

Reviewer #3

Reviewer 3 feels that the authors did a commendable job of addressing the concerns articulated and believes that the manuscript is now suitable for publication in Nature Communications. Some matters still remain unresolved (e.g., the mechanism underlying the induction of innate immune activation [presumably mediated by peptidoglycans, based on the trypsin data, though the specific pattern recognition receptor remains undefined] -- which is likely the most interesting aspect of this project); still, the conclusions are generally well supported by the data presented, and the findings appear to be quite robust.

- We thank the reviewer for the positive comment regarding our revised manuscript.
- We deeply agree with the reviewer that studies regarding the precise mechanism underlying the induction of immune activation by bacterial extracellular vesicles are of great importance in various aspects. We are carrying out further studies to reveal the specific bacterial components or pattern recognition receptors responsible for bacterial extracellular vesicle-mediated antitumor effect, and hope to publish our new results in the near future.